# SHEP: Spatial Heterogeneity–Driven Experience Prioritization in Scalable Multi-Agent Reinforcement Learning

## Abstract

Scalable Multi-Agent Reinforcement Learning (MARL) faces severe challenges regarding the exponential explosion of joint state-action space dimensionality and the difficulty of global coordination as the number of agents increases. Traditional methods optimize fine-grained individual strategies within an exponentially vast state space, leading to low sample efficiency and training bottlenecks in large-scale scenarios. To address these issues, this paper proposes SHEP (Spatial Heterogeneity–Driven Experience Prioritization), a mesoscopic guidance framework designed for large-scale group coordination. SHEP utilizes Occupancy Entropy, Action Diversity Entropy, and Moran's I to construct a set of topological feature descriptors, mapping the high-dimensional individual state space into a low-dimensional, interpretable group feature space. Building on this, we design heterogeneity-driven prioritized experience replay and Group Hindsight Experience Replay (Group-HER). By identifying critical moments of abrupt spatial heterogeneity changes or highly structured clustering, these mechanisms accurately screen for high-value samples and perform "dimensionality reduction pruning" on the ineffective exploration space, significantly improving sample efficiency. Due to the universality of its experience screening mechanism, SHEP can be seamlessly integrated as a "plug-in" into mainstream centralized training algorithms like MAPPO without altering their underlying policy optimization objectives. In MAgent environments and SMAC benchmarks, SHEP demonstrates superior performance, with convergence speed and final win rates significantly outperforming baseline methods such as QMIX and Mean-Field approaches. These results robustly validate that introducing explicit spatial heterogeneity features to guide experience prioritization is an effective paradigm for resolving the curse of dimensionality in scalable MARL.

## 1 Introduction

In the frontier of modern artificial intelligence applications, Multi-Agent Systems (MAS) are gradually transitioning from theoretical simulations to complex real-world deployments (Ma et al., 2024). Tasks such as cooperative navigation of drone swarms, automated handling in warehousing logistics, and joint search in disaster rescue operations typically involve high dynamics and complex spatiotemporal constraints (Hu et al., 2020). To address these challenges, the field of Multi-Agent Reinforcement Learning (MARL) has seen the emergence of a series of classic algorithms, including value decomposition-based methods like QMIX (Rashid et al., 2020) and QPLEX, as well as policy gradient-based methods like MAPPO (Yu et al., 2022). These approaches have achieved remarkable performance in small-to-medium-scale collaborative tasks by refining the modeling of joint states and credit assignment among agents.

However, as task scenarios expand to large-scale clusters with tens or even hundreds of agents, the dimensionality of the joint state-action space exhibits an exponential explosion, presenting severe scalability bottlenecks for traditional MARL methods (Liu et al., 2024; Zhang et al., 2021). To mitigate this curse of dimensionality, researchers have conducted extensive explorations. For instance, Graph Neural Network (GNN)-based methods attempt to reduce interaction complexity by sparsifying communication topologies (Fu et al., 2022), while Mean-Field Reinforcement Learning (MF-MARL) approximates joint policies by

simplifying agent interactions into binary games between individuals and the average effect of their neighbors (Yang et al., 2018; Cui et al., 2024). Fundamentally, these existing studies primarily focus on patching and improving existing frameworks. Their core remains limited to a "bottom-up" optimization paradigm, attempting to find optimal fine-grained strategies for each individual directly within an extremely vast state space. This paradigm relies on massive trial-and-error in hopes that macroscopic intelligence will automatically emerge from microscopic strategies, which proves extremely inefficient in the context of vast solution spaces and non-stationary environments.

We argue that large-scale multi-agent systems are not merely simple superpositions of small-scale systems; the quantitative change in scale induces a qualitative change in system characteristics (Ning & Xie, 2024). In large-scale group tasks, the marginal contribution of a single agent's microscopic action to the global outcome is significantly reduced, while the macroscopic behavioral patterns of the group become the key factors determining task progression. To leverage this characteristic, some methods based on macroscopic instructions or abstract hierarchies attempt to introduce high-level guidance (Tang et al., 2018). Such approaches typically employ a hierarchical architecture where a high-level policy formulates macroscopic instructions (e.g., "attack area A" or "maintain formation"), and low-level policies are responsible for specific execution. While this explicit hierarchical division simplifies training and decision logic, it faces significant limitations. First, macroscopic variables are often set as domain-dependent latent variables or predefined rules, making generalization difficult. Second, training high-level policies often requires introducing additional sparse rewards or complex multi-level optimization, leading to increased training instability. Furthermore, rigid hierarchical segmentation may result in a disconnect between "instruction" and "execution," where macroscopic instructions fail to flexibly adapt to the dynamic physical constraints of underlying individuals, leading to rigid final strategies that struggle to cope with highly dynamic adversarial environments.

This leads to a core question: Can we construct a mechanism that flexibly guides the efficient evolution of microscopic strategies using macroscopic spatial features of group behavior, without introducing rigid hierarchical instructions? The core lies in establishing a benign "macro-micro" mapping: we do not need a coercive commander to replace individual microscopic control, but rather a "evaluator" with a macroscopic perspective to inform individuals which spatial coordination forms are valuable.

This paper proposes the SHEP (Spatial Heterogeneity–Driven Experience Prioritization) framework, aiming to solve the collaborative learning difficulties in the aforementioned large-scale MARL scenarios. SHEP starts from physical principles to construct a mesoscopic metric system based on Spatial Heterogeneity. We utilize topological statistics such as Occupancy Entropy and Moran's I (Moran, 1950) to quantify the dynamic balance of the group between "global dispersion" and "local concentration"—that is, maximizing spatial coverage macroscopically while forming tight structured collaboration locally to establish advantages. Centered on this core idea, SHEP designs a spatial heterogeneity-driven experience screening mechanism that prioritizes replays of trajectory segments exhibiting benign spatial configuration evolution, thereby greatly compressing the ineffective exploration space. As a general methodological paradigm, SHEP can be seamlessly integrated on top of mainstream CTDE algorithms like MAPPO. In large-scale adversarial experiments on MAgent and SMAC, SHEP not only leads significantly in convergence speed but also demonstrates excellent robustness in complex terrains and force-disadvantage scenarios, proving that mesoscopic guidance based on spatial heterogeneity is an effective pathway to achieving the emergence of large-scale group intelligence.

## 2 Related Work

### 2.1 Bottom-Up Coordination based on Micro-Interactions

In recent years, mainstream Multi-Agent Reinforcement Learning methods have followed the Centralized Training with Decentralized Execution (CTDE) paradigm (Wen et al., 2021). This category covers policy gradient methods with centralized critics, such as MADDPG and MAPPO (Yu et al., 2022), as well as value decomposition-based methods like VDN and QMIX (Rashid et al., 2020). These methods allow agents to act based on local observations while utilizing global information during training to learn optimal collaboration. For example, QMIX decomposes the joint Q-value function into agent-specific values with monotonicity constraints, effectively extracting decentralized policies, and has achieved significant results on tasks such as

*StarCraft II*. However, as group size increases, the limitations of such methods become apparent: the size of the joint action space grows exponentially, causing a sharp rise in sample complexity, and the centralized critic itself becomes massive and difficult to train (Zhang et al., 2021).

To address the curse of dimensionality brought by scale expansion, the research community has proposed various approximate interaction paradigms based on the aforementioned frameworks. For instance, Mean-Field Reinforcement Learning (Mean-Field MARL) draws on ideas from statistical physics to simplify agent-group interactions into binary games between individuals and the average effect of their neighbors (Yang et al., 2018; Cui et al., 2024). Subsequent work introduced Graph Neural Networks (GNNs) and attention mechanisms to capture key interactions by learning sparse communication topologies (Fu et al., 2022). Although these methods support large-scale agents at the computational level, their essence remains "bottom-up" microscopic optimization. The algorithms still require massive samples to converge, limiting their practical performance ceiling in complex large-scale tasks (Xu et al., 2024).

## 2.2 Top-Down Macro-Guidance based on Hierarchical Architectures

Another research path explicitly introduces hierarchical structures in MARL to focus on macroscopic behaviors, adopting a top-down perspective. Such methods typically divide policies into different levels, for instance, using a high-level controller to select macroscopic instructions (e.g., move to a specific area) and a low-level controller for specific execution (Tang et al., 2018). While this hierarchical design helps with credit assignment in long-horizon tasks, its main limitation lies in its high dependence on domain knowledge. Existing hierarchical methods often require predefining fixed hierarchical structures or introducing additional supervisory signals to train high-level policies, which largely restricts the algorithm's generalization ability and flexibility in non-specific tasks. Moreover, rigid hierarchical segmentation may lead to a disconnect between "instruction" and "execution," making it difficult to cope with highly dynamic gaming environments.

## 2.3 Experience Replay and General Screening Mechanisms

In addition to optimizing policy structure, improving data utilization efficiency is another critical paradigm in reinforcement learning. Since the spatial heterogeneity metrics proposed in this paper possess physical universality, they can serve as a general basis for experience screening, allowing this method to be seamlessly integrated into various existing frameworks as a "plug-in." The idea of screening high-value samples through designing general metrics to improve sample efficiency has been widely validated in Single-Agent Reinforcement Learning (Single-Agent RL). Representative works include Prioritized Experience Replay (PER) (Schaul et al., 2016) and Hindsight Experience Replay (HER) (Andrychowicz et al., 2017).

PER measures the learning value of samples based on Temporal-Difference Error (TD-Error). Its core hypothesis is that samples with larger TD errors often contain more "unexpected" information and should therefore be assigned higher sampling priority. In this way, PER can replay experiences not yet mastered by the current policy more frequently, significantly accelerating convergence. HER addresses the sparse reward problem by proposing an ingenious relabeling mechanism: when an agent fails to achieve a predetermined goal, the algorithm retroactively labels the final state reached as a "virtual goal," thereby transforming failed trajectories into successful training samples. This method greatly improves data utilization in goal-oriented tasks.

However, the aforementioned methods are primarily limited to single-agent domains or specific types of tasks. When directly applying PER to Scalable Multi-Agent Reinforcement Learning (MARL), TD errors are often full of noise due to environmental non-stationarity, making it difficult to truthfully reflect sample value (Papoudakis et al., 2021). HER, on the other hand, relies on clear goal state definitions, which are difficult to transfer directly to complex adversarial group games. Recent works have attempted to adapt these mechanisms to multi-agent settings, such as AccMER (Gogineni et al., 2023) and MAC-PO (Mei et al., 2023), which focus on cache locality or collective priority optimization. In contrast, the SHEP framework proposed in this paper is the first to utilize "spatial heterogeneity" features with physical universality as screening criteria, providing a general experience prioritization and relabeling mechanism for scalable MARL without altering the underlying algorithm logic.

## 3 Preliminaries

We model the collaborative task of a large-scale multi-agent system as a Decentralized Partially Observable Markov Decision Process (Dec-POMDP) (Oliehoek & Amato, 2016). Formally, this process is defined by a tuple $\langle \mathcal{S}, \mathcal{A}, \mathcal{O}, N, P, R, \gamma \rangle$, where $N$ represents the number of agents. At each time step $t$, the system is in a global state $s_t \in \mathcal{S}$. Each agent $i \in \{1, \ldots, N\}$ receives its local observation $o_t^i \in \mathcal{O}$ and selects an action $a_t^i \in \mathcal{A}$ according to its policy $\pi^i(a_t^i | o_t^i)$, forming a joint action $\boldsymbol{a}_t = \{a_t^1, \ldots, a_t^N\}$. The environment evolves according to the state transition function $P(s_{t+1} | s_t, \boldsymbol{a}_t)$, and all agents share a global team reward $r_t = R(s_t, \boldsymbol{a}_t)$. The system's goal is to learn a joint policy $\boldsymbol{\pi}$ to maximize the expected discounted cumulative return $J(\boldsymbol{\pi}) = \mathbb{E}_{\boldsymbol{\pi}}[\sum_{t=0}^{\infty} \gamma^t r_t]$.

To address non-stationarity in large-scale environments, we adopt the Centralized Training with Decentralized Execution (CTDE) paradigm (Yu et al., 2022). Within this framework, the learning process allows access to the global state $s_t$ to train a centralized Critic to guide the update of the Actor; during the execution phase, each agent relies solely on local observations $o_t^i$ for decentralized decision-making. Although CTDE performs excellently in small-to-medium-scale tasks, when the number of agents expands to the scale of hundreds, the exponential expansion of the joint state-action space leads to extreme credit assignment difficulties and sample inefficiency. This is precisely the core issue the SHEP framework proposed in this paper strives to resolve.

## 4 Methodology

Addressing the challenges of joint state space explosion and sparse effective coordination signals in Large-Scale Multi-Agent Systems (Large-Scale MAS), this paper proposes the SHEP (Spatial Heterogeneity-Driven Experience Prioritization) framework. In complex systems involving hundreds or thousands of agents, attempting to have a global optimal strategy "emerge" directly from massive interactions of microscopic individuals is extremely inefficient (Xu et al., 2024). The core idea of SHEP lies in introducing a "Mesoscopic" perspective, utilizing spatial heterogeneity features with physical semantics to bridge microscopic actions and macroscopic task goals. By quantifying the orderliness and diversity of the group's spatiotemporal distribution, SHEP can accurately identify "critical coordination segments" with structured features (e.g., efficient coverage, tight collaborative clustering) from massive and noisy exploration data. It then guides the algorithm to prioritize learning these high-value experiences, thereby significantly improving sample efficiency in large-scale scenarios (Kim et al., 2023). The overall architecture and execution flow of the SHEP framework are shown in Figure 1. Its core logic consists of three tightly coupled stages:

**Mesoscopic Situation Assessment**: During the environmental interaction phase, Module A calculates the group's spatial distribution entropy and topological coordination index in real-time, mapping high-dimensional microscopic states to low-dimensional mesoscopic situation scores.

**Semantic-Driven Sample Prioritization**: Based on situation assessment results, Module B (Group-HER) backtracks historical trajectories to identify and label "potential success" segments with tactical value; Module C (Prioritized Replay) calculates trajectory-level priorities based on the magnitude of spatial heterogeneity evolution, constructing a mesoscopic-aware experience replay buffer.

**Unbiased Weighted Optimization**: Utilizing Importance Sampling weights to correct data distribution bias, this stage guides the Base Learner to update policies, ensuring the final policy converges to the optimal solution.

### 4.1 Scale-Invariant Mesoscopic Construction

In large-scale MAS, the number of agents $N$ often varies dynamically, and directly using an $N$-dimensional joint state leads to linear or even exponential growth in input dimensionality with scale. To capture group behavioral patterns robust to scale changes (Scale-Invariant Patterns), we need to construct a set of mesoscopic representation bases independent of the specific number of agents.

**Density Field Projection Mechanism:**

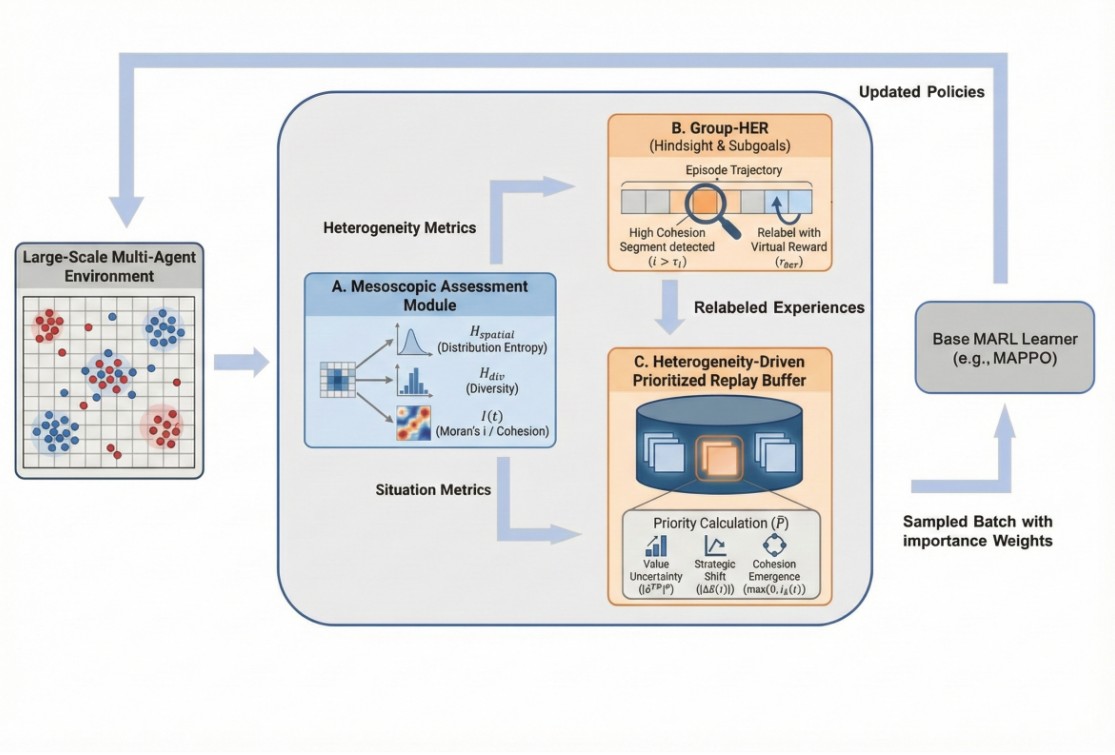

Figure 1: Overview of the SHEP Framework

We introduce a fixed reference frame independent of the number of agents. Specifically, based on a task-specific metric space $(\mathcal{X}, d)$, we partition the environment into $M$ non-overlapping "Macro-cells" $\mathcal{C} = \{C_1, \ldots, C_M\}$.

In this framework, the metric function $d : \mathcal{X} \times \mathcal{X} \to \mathbb{R}_+$ determines the spatial topological structure. This metric possesses high universality: for instance, in physical space collaboration tasks, $d$ is typically instantiated as Euclidean distance, while in communication networks or logistics topologies, it can manifest as geodesic distance on a graph. Regardless of the specific form of $d$, its core role is to induce adjacency relationships in the environment and establish rules for cell partitioning.

At each time step $t$, the algorithm projects the microscopic states of all agents into the aforementioned macro-cells and calculates the relative density of agents falling into each cell $p_i(t) = n_i(t)/N$, thereby generating a density field vector $P_t = [p_1, \ldots, p_M]$.

This projection mechanism achieves scale invariance on two levels:

1. **Constancy of Feature Dimensionality**: Regardless of how the number of agents $N$ fluctuates in the environment, the dimension of the density field $P_t$ remains fixed at $M$. This means we fundamentally shift the description of coordination patterns from "individual quantity dependence" to "spatial topology dependence."

2. **Generalization of Coordination Patterns**: The density field focuses on the relative morphology of group distribution (e.g., "dispersed coverage" or "local aggregation") rather than absolute numbers. This normalized representation enables the algorithm to capture general collaboration structures across scales.

Simultaneously, since the number of macro-cells $M$ is a preset constant designed to satisfy $M \ll N$, subsequent spatial statistical calculations rely only on the scale of $M$. Therefore, no matter how the agent

scale $N$ grows, the computational complexity of feature extraction remains at the $O(M)$ level, theoretically guaranteeing the algorithm's computational efficiency during large-scale expansion.

## 4.2 Quantification of Spatial Heterogeneity

Based on the aforementioned density field $P_t$, we further define a set of statistical descriptors to quantify the spatial heterogeneity of the group from two dimensions: "distribution breadth" and "structural depth." These metrics constitute the value criteria for subsequent experience screening.

**1. Occupancy Entropy ($H_{occ}$)**

To quantify the degree of environmental exploration and resource coverage efficiency of the group, we use information entropy to define Occupancy Entropy:

$$H_{occ}(t) = -\sum_{i=1}^{M} p_i(t) \log p_i(t) \tag{1}$$

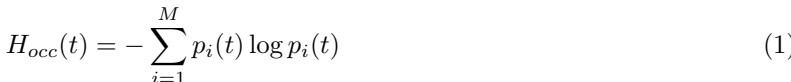

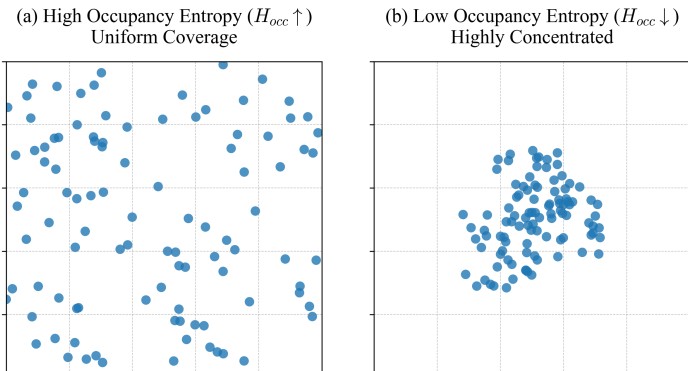

Figure 2: Illustration of Occupancy Entropy

The physical meaning of $H_{occ}$ is intuitive and clear: a high entropy value indicates that agents have achieved maximized coverage within the reachable space, corresponding to efficient global exploration or multi-task parallel processing capabilities; a low entropy value implies a highly concentrated group distribution. This provides the algorithm with a general objective metric for assessing "spatial utilization efficiency."

**2. Topological Coordination Index / Moran's I**

Distribution breadth alone is insufficient to describe complex collaboration. We introduce Moran's I (Moran, 1950) from spatial statistics to quantify the spatial autocorrelation of the density field, i.e., "structural depth." Let $z_i(t)$ be the standardized density of cell $i$, and $w_{ij}$ be the spatial adjacency weight induced by metric $d$, then:

$$I(t) = \frac{M}{\sum_{i,j} w_{ij}} \frac{\sum_{i,j} w_{ij}(z_i(t) - \bar{z})(z_j(t) - \bar{z})}{\sum_i (z_i(t) - \bar{z})^2} \tag{2}$$

Moran's I can keenly distinguish between "random scattering" and "structured aggregation." $I(t) > 0$ indicates that high-density regions are adjacent to each other in topology, revealing that the group has formed tight collaborative clustering or continuous structures; $I(t) \approx 0$ corresponds to disordered random distribution. This indicator enables the algorithm to capture the spontaneous emergence of local collaborative behaviors.

In addition, we briefly examine Action Diversity Entropy $H_{div}(t)$ as an auxiliary indicator to monitor the richness of group behavioral patterns and prevent Mode Collapse. The above indicators jointly constitute

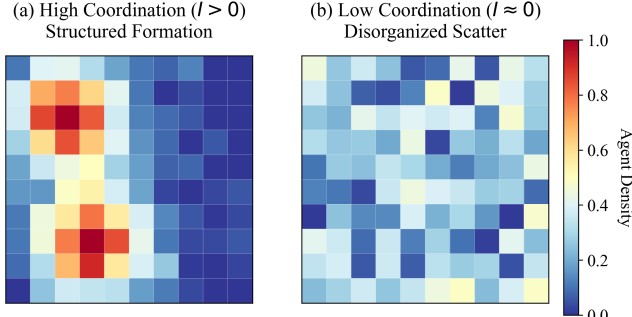

Figure 3: Illustration of Moran's I

the Spatial Heterogeneity Descriptors (SHD), concretizing abstract "group coordination" into computable physical statistics.

### 4.3 Heterogeneity-Driven Prioritized Experience Replay

In large-scale collaborative tasks, sparse global rewards often lead to severe sample efficiency problems: agents may undergo lengthy exploration to achieve a complex collaborative goal once, but standard random sampling replay struggles to highlight this critical success path amidst massive amounts of ineffective data (Albrecht & Stone, 2018). To this end, we propose an experience screening mechanism based on spatial heterogeneity. Its core lies in resolving two key issues: Good spatial configurations (such as effective coverage or tight collaborative formations) are often necessary preconditions for task success. By using mesoscopic metrics as screening criteria, the algorithm can leverage trajectory segments showing correct coordination trends before receiving final rewards. Simultaneously, unlike purely numerical error metrics, spatial heterogeneity directly reflects coordination quality at the physical level (Li et al., 2024). This enables the algorithm to distinguish between "meaningless high-frequency oscillations" and "meaningful situational transitions," thereby guiding the model to focus on structured behaviors that truly drive situational changes.

To translate the above ideas into an executable algorithm flow, we designed a complete method ranging from metric quantification to unbiased sampling:

#### 4.3.1 Construction of Spatial Heterogeneity Advantage Metrics

To transform the high-dimensional SHD vector $\mathbf{\Phi}_t$ into a scalar signal usable for ranking, we first define the comprehensive group score $E(t)$. This score fuses information from spatial coverage, action diversity, and structural coordination through linear weighting:

$$E(t) = \alpha H_{\text{occ}}(t) + \beta H_{\text{div}}(t) + \eta I(t) \tag{3}$$

where $\alpha, \beta, \eta$ are hyperparameters balancing the contribution of each dimension. Based on this, we further define the relative advantage $\delta(t)$ to quantify the degree of dominance of our group relative to the opponent at the current moment:

$$\delta(t) = \kappa[E_{\text{self}}(t) - E_{\text{opp}}(t)] \tag{4}$$

Compared to sparse win/loss rewards returned by the environment (usually given only at the end of an episode), $\delta(t)$ provides a temporally dense macroscopic proxy signal. It can keenly capture small but critical cumulative changes in the battle situation—such as increased spatial control brought by formation deployment, or a situational tilt caused by a successful local encirclement.

SHEP introduces a hybrid semantic prioritization mechanism. We combine traditional "microscopic level" priority (TD error) with two "macroscopic level" components derived from mesoscopic metrics. This reflects our intuition regarding what constitutes valuable learning events: sudden changes in mesoscopic advantage, and the formation of coherent clusters (a hallmark of effective group collaborative structures). Specifically, for each time step $t$ in an episode (for each environment instance), we define the unnormalized priority score $\widetilde{P}(s_t, a_t)$ as:

$$\widetilde{P}(s_t, a_t) = \underbrace{|\delta_t^{\mathrm{TD}}|^\alpha}_{\text{Value Uncertainty}} + \underbrace{\lambda_\Delta |\Delta\delta(t)|}_{\text{Strategic Shift}} + \underbrace{\lambda_I \max\{0, I_A(t)\}}_{\text{Cohesion Emergence}} \tag{5}$$

The meanings of the symbols are as follows:

- $\left|\delta_t^{\mathrm{TD}}\right|^\alpha$ represents the absolute value of the TD error at time $t$ (for the current joint action), raised to the power of $\alpha$ (a hyperparameter controlling the intensity of preference for large error samples, typically set to $\alpha = 0.6$ as in PER (Schaul et al., 2016)). This term corresponds to traditional priority, characterizing the uncertainty or "surprise" of the value function. We retain this term so that the base algorithm can still focus on samples that are difficult to predict and thus have high learning potential.

- $|\Delta\delta(t)|$ describes the absolute magnitude of change in mesoscopic advantage $\delta(t)$ relative to the previous time step (or a short time window). A large $\Delta\delta$ indicates a drastic turn in the battle momentum at time $t$, such as a large number of units being destroyed in a short time, or one side suddenly occupying a key strategic position. By adding the term $\lambda_\Delta |\Delta\delta(t)|$ weighted by coefficient $\lambda_\Delta$, we explicitly increase the priority of experiences near these turning points, guiding the learner to focus on key factors leading to sudden advantage changes.

- $\max(0, I_A(t))$ is the non-negative part of the Moran's I of our group at time $t$. We scale this metric by coefficient $\lambda_I$ to boost the priority of these samples when the group exhibits positive spatial correlation (i.e., when forming clusters or collaborative formations, as $I_A(t) > 0$ implies structured aggregation). We ignore negative $I$ values because disordered dispersion does not necessarily imply meaningful tactical patterns and is often just noise. Emphasizing positive $I$ is equivalent to assigning greater weight to samples where agents are clearly aggregated in space and likely to engage in high-intensity collaborative actions, thereby helping to filter out transitions where agents are randomly dispersed and contribute little to learning excellent collaborative strategies.

Through the combination of these three components, we obtain a more semantic experience ranking mechanism: prioritizing moments with large value estimation deviations, accompanied by major strategic changes, or exhibiting significant spatial structures (Saglam et al., 2023). By adjusting hyperparameters $\lambda_\Delta$ and $\lambda_I$, we can control the extent to which we emphasize these macroscopic events relative to the original TD error. In experiments, we observed that this hybrid priority significantly accelerated the learning process because the replay buffer was no longer flooded with a large number of trivial or high-noise transitions.

**Trajectory-Level Priority Ranking:** Instead of sampling individual transitions with probability proportional to $\widetilde{P}$, we perform trajectory-level priority ranking. Multi-agent tasks have strong temporal correlation, and critical moments usually span several continuous steps (e.g., engagement sequences). Therefore, we first segment experiences into trajectories (or episodes, or key segments) and assign an average priority $\bar{P}_i = \mathbb{E}_t\left[\widetilde{P}(s_t, a_t)\right]$ to each trajectory $i$ over its time steps. This averaging eliminates single-step peaks and identifies trajectories that contain high strategic value as a whole. Then, we normalize these priorities and obtain a sampling probability for each trajectory $i$:

$$p_i = \frac{\bar{P}_i}{\sum_{j=1}^{N_{\text{buffer}}} \bar{P}_j} \tag{6}$$

where the summation is performed over all available trajectories in the replay buffer (or current batch). We also clip excessively large $p_i$ to avoid any single trajectory dominating sampling. In this way, sampling frequency is automatically skewed towards episodes with higher information density and more critical events (Liu et al., 2025).

**Unbiased Importance Sampling Correction:** Adopting prioritized sampling changes the data distribution, theoretically introducing bias into the learning process, which may cause the convergence point to deviate from the true optimal solution. To this end, similar to PER, we apply Importance Sampling (IS) weights to the loss of each sampled trajectory. If trajectory $i$ is sampled with probability $p_i$, its IS weight is defined as:

$$w_i = \left( \frac{1}{N_{\text{batch}} \cdot p_i} \right)^{\beta} \tag{7}$$

where $N_{\text{batch}}$ is the batch size (number of sampled trajectories), and $\beta$ is an annealing factor that starts from a small value (to avoid over-correction early on when priorities may be noisy) and gradually increases to 1. The loss of the trajectory (e.g., policy gradient or TD loss aggregated over it) is multiplied by $w_i$. Intuitively, if a trajectory is sampled too frequently (high $p_i$), we reduce the weight of its update; if it is sampled rarely, we increase its weight.

When $\beta \to 1$, this method ensures that the expectation of the weighted gradient is consistent with the gradient expectation under uniform experience sampling, thereby restoring unbiasedness. The practical effect is that SHEP only alters the speed and path of optimization (by focusing on more informative data earlier) without changing the final convergence objective; the optimal solution still corresponds to the original objective function. Specifically, when $\beta = 1$:

$$\mathbb{E}_{i \sim p} \left[ w_i \nabla L_i \right] = \mathbb{E}_{i \sim \text{uniform}} \left[ \nabla L_i \right] \tag{8}$$

Thus, on a sufficiently long time scale, SHEP updates are equivalent to standard uniform sampling updates, only achieving convergence more efficiently.

Leveraging this mesoscopic-aware prioritized experience replay mechanism, we can concentrate learning resources on truly critical experience segments, such as large-scale battles or successful formation charges, while filtering out samples that appear repeatedly in large-scale training but are information-poor. This not only significantly improves sample efficiency but also helps agents abstract effective patterns from the results of key strategic decisions more quickly.

To further improve the robustness of the algorithm under extremely sparse rewards, SHEP also utilizes the aforementioned mesoscopic metrics to construct auxiliary exploration and relabeling mechanisms. Specifically, we introduce intrinsic rewards based on mesoscopic feedback, giving immediate feedback for behaviors that improve spatial entropy and coordination index in the early training stages, guiding agents to quickly master basic formations via a smooth soft curriculum. Additionally, for failed trajectories, we design Group Hindsight Experience Replay (Group-HER) (Zeng et al., 2023). By backtracking to detect "high coordination segments" or "moments of situational reversal" within the trajectory, we label them as virtual sub-goals and assign success rewards. These auxiliary mechanisms complement the core prioritized sampling, further mining potential structured signals in the high-dimensional exploration space.

Algorithm 1 fully demonstrates the execution flow of the SHEP framework (see Appendix A.1).

## 5 Experiments

To systematically evaluate the scalability and sample efficiency of the SHEP framework, we conducted phased empirical studies on two mainstream benchmark platforms: MAgent (Zheng et al., 2018) and StarCraft II (SMAC) (Samvelyan et al., 2019). We selected MAPPO (Yu et al., 2022), QMIX (Rashid et al., 2020), and MFAC (Cui et al., 2024) as baselines. All experiments were run independently with 5 random seeds.

### 5.1 MAgent Large-Scale Adversarial Scenarios

On the MAgent platform, we constructed adversarial environments of two scales (49v49 and 100v100) with "Open-Area" and "Fixed-Obstacle" maps (Figure 4). Opponent policies were trained to convergence using MAPPO via Self-Play.

In the **Open-Area** battlefield, SHEP demonstrated superior efficiency. In 100v100, both SHEP and MFAC rapidly increased win rates, indicating that density-based biases effectively guide deployment in unconstrained spaces. However, in 49v49, MFAC degraded later, while SHEP maintained stability. This stems from SHEP constraining exploration to a mesoscopic manifold with physical semantics (SHD), avoiding the "high-dimensional meaningless random walks" common in large joint spaces (Liu et al., 2024).

When **Fixed-Obstacle** constraints were introduced, SHEP's advantages became decisive (Figure 5). MFAC suffered from oscillations and lower convergence. This profoundly reveals the limitation of mean-field assumptions: homogenizing neighborhood behaviors causes ineffective congestion at narrow bottlenecks. In contrast, SHEP stabilized at 100% win rate by explicitly rewarding structured positive correlations via Moran's I (Moran, 1950). This topological guidance acts as substantial "dimensionality reduction pruning" (Fu et al., 2022), prompting agents to spontaneously emerge high-order patterns compatible with the environmental topology.

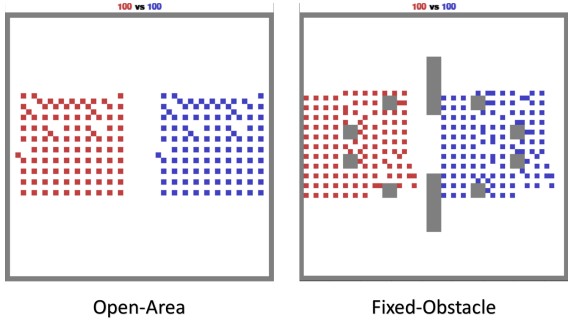

Figure 4: MAgent Experimental Scenarios

### 5.2 SMAC Cooperative Adversarial Scenarios

We further transferred the method to the SMAC platform to evaluate robustness in high-dynamic micromanagement. We focused on the large-scale **27m_vs_30m** scenario (3M steps) and other typical maps (1M steps).

In the **27m_vs_30m** scenario (Figure 6), SHEP maintained stable improvement, achieving ≈80% win rate, significantly outperforming QMIX (≈55%) and MAPPO (≈20%). Since this scenario involves large formations and force disadvantage, results show that standard CTDE struggles with credit assignment for transient high-value segments. SHEP, guided by mesoscopic situations, effectively captures these local advantages, achieving counter-trend learning.

In other scenarios: (1) **25m**: Baseline MAPPO failed to learn due to exploration collapse (Hao et al., 2023). SHEP used intrinsic rewards (occupancy entropy) to obtain dense gradient signals, reaching >40% win rate in ≈100k steps and stabilizing above 60%, an order-of-magnitude efficiency gain. (2) **5z**: SHEP's steeper learning curve indicates situational guidance rapidly catalyzes tactical patterns. (3) **8m_vs_9m**: Baselines failed (≈0% win rate) due to severe disadvantage. SHEP reached ≈50% win rate by extracting tactical structures from failed samples via Group-HER (Zeng et al., 2023).

Unlike in MAgent, MFAC provided limited benefits in SMAC. Simple neighborhood averaging obscures the fine-grained micromanagement details required in StarCraft II (Hao, 2022). SHEP, by explicitly modeling structural changes through topological descriptors, demonstrates superior cross-platform robustness. Even

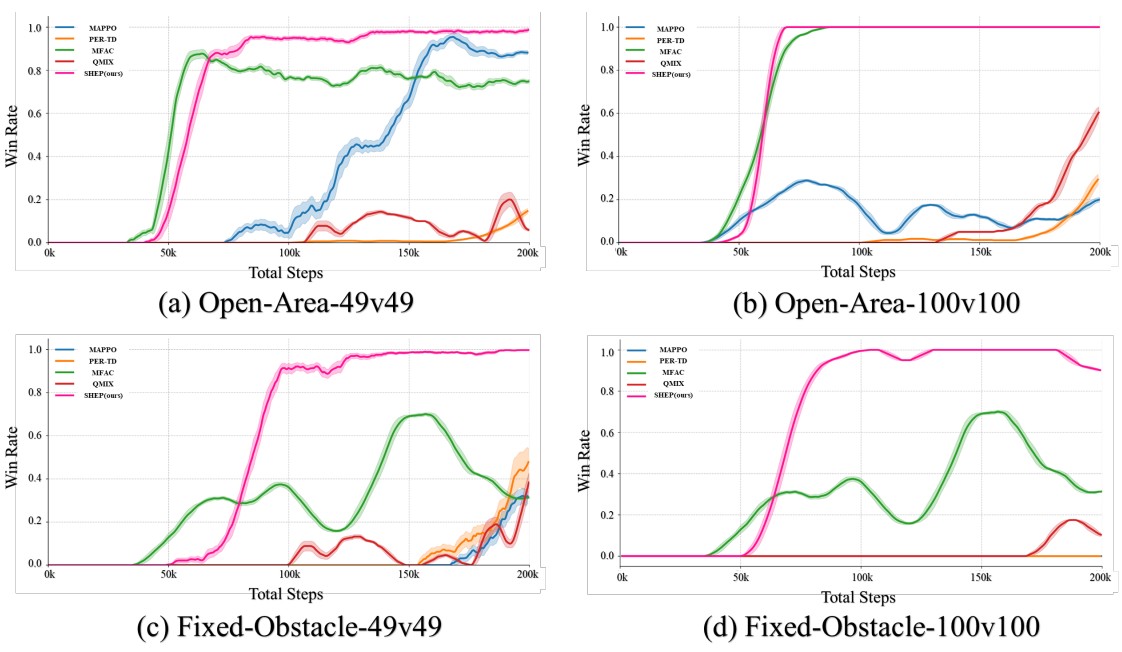

Figure 5: Performance comparison in MAgent scenarios

in defeat, SHEP distills transient advantages into "local successful experiences" for policy updates through replay relabeling, constructing an effective automatic curriculum.

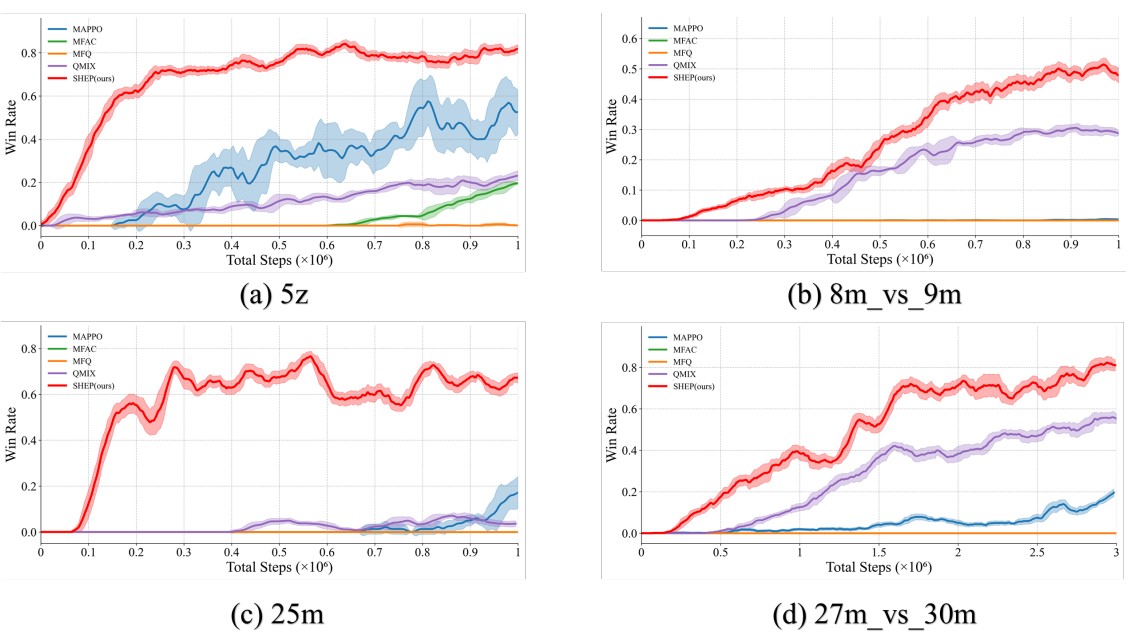

Figure 6: Performance comparison in SMAC scenarios

# 6 Conclusions and Discussions

Addressing the low sample efficiency in Scalable MARL, this paper proposes SHEP. Unlike methods that simplify interaction models, SHEP constructs Spatial Heterogeneity Descriptors (Occupancy Entropy and Moran's I) to map high-dimensional microscopic states to semantically clear mesoscopic features. Centered on this, we integrate heterogeneity-driven prioritized replay and Group-HER as "mesoscopic plug-ins," performing substantial "dimensionality reduction pruning" on the vast exploration space. Experimental results on MAgent and SMAC demonstrate that SHEP significantly outperforms baselines such as MAPPO and MFAC in convergence speed and robustness, particularly in complex topologies and disadvantageous scenarios. These results validate that explicit mesoscopic guidance effectively mitigates the curse of dimensionality without sacrificing microscopic control precision (Fabian et al., 2023).

Examining the operating mechanism, SHEP succeeds by reshaping learning into an "implicit curriculum": early intrinsic rewards for spatial orderliness naturally transition to functional optimization, simulating human command logic to reduce exploration difficulty. Furthermore, the mesoscopic-aware prioritization acts as a "semantic filter" against non-stationarity. Unlike traditional sampling which is easily disturbed by noisy TD errors (Papoudakis et al., 2021), SHEP weights experiences by topological significance, ignoring high-variance noise to focus on substantive situational turns (Fan et al., 2020).

A current limitation is the dependence on the metric function. While experiments primarily used Euclidean distance, SHEP's theoretical framework is metric-agnostic. Future work will explore adaptive metric learning mechanisms, such as automatically constructing latent distances via causal inference, to enhance SHEP's generalization in non-Euclidean and broad abstract collaborative tasks (Liu et al., 2024).

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

# A  Appendix

## A.1  SHEP Algorithm Pseudocode

---

**Algorithm 1** SHEP: Spatial Heterogeneity-Driven Experience Prioritization

---

**Require:** Batch of trajectories $\mathcal{T}$ collected by base policy (e.g., MAPPO)
**Require:** Hyperparameters: $\alpha, \beta, \eta$ (metric weights), $\lambda_\Delta, \lambda_I$ (priority weights)
1: **Input:** Initialized Replay Buffer $\mathcal{D}$
2: **for** each trajectory $\tau \in \mathcal{T}$ **do**
3:     // **1. Mesoscopic Situation Assessment & Intrinsic Reward**
4:     **for** each timestep $t$ in $\tau$ **do**
5:         Compute Metrics: $H_{occ}(t)$ (Coverage), $H_{div}(t)$ (Diversity), $I(t)$ (Cohesion)
6:         Situation Score: $E(t) \leftarrow \alpha H_{occ}(t) + \beta H_{div}(t) + \eta I(t)$
7:         Calculate Advantage: $\delta(t) \leftarrow \kappa[E_{self}(t) - E_{opp}(t)]$
8:         Intrinsic Reward: $r_t^{int} \leftarrow \beta_t \Delta E(t) + \eta_t I(t)$
9:         Augment Reward: $r_t^{total} \leftarrow r_t^{env} + r_t^{int}$
10:    **end for**
11:    // **2. Group-HER (Group Hindsight Experience Replay)**
12:    **if** segment in $\tau$ shows High Cohesion ($I(t) > \tau_{th}$) $\vee$ Advantage Rise ($\Delta\delta(t) > 0$) **then**
13:        Create trajectory copy $\tau_{her}$
14:        Relabel transitions in $\tau_{her}$ with virtual success reward $r_{her}$
15:        Add $\tau_{her}$ to Replay Buffer $\mathcal{D}$
16:    **end if**
17:    // **3. Heterogeneity-Driven Priority Calculation**
18:    Compute TD-Error $|\delta_t^{TD}|$ for all $t$ using Critic
19:    Calculate Semantic Priority $\tilde{P}(s_t, a_t)$ for each step:
20:        $\tilde{P}(s_t, a_t) \leftarrow \underbrace{|\delta_t^{TD}|^\alpha}_{\text{Value Uncertainty}} \cdot \underbrace{(\lambda_\Delta|\Delta\delta(t)| + \lambda_I \max\{0, I(t)\})}_{\text{Tactical Significance}}$
21:    Compute Trajectory Priority: $\bar{P}_\tau \leftarrow \mathbb{E}_t[\tilde{P}(s_t, a_t)]$
22:    Store $\tau$ in $\mathcal{D}$ with priority $\bar{P}_\tau$
23: **end for**
24: // **4. Unbiased Weighted Optimization**
25: Sample batch $B \sim \bar{P}$ from $\mathcal{D}$
26: Compute IS weights: $w_i = (N_{batch} \cdot p_i)^{-\beta}$
27: Update Policy $\pi_\theta$ and Critic $V_\phi$ using Weighted Loss:
28:    $\mathcal{L} = \mathbb{E}_B[w_i \cdot \mathcal{L}_{Base}(\tau)]$

---

