# OpenReview forum: "SHEP: Spatial Heterogeneity–Driven Experience Prioritization in Scalable Multi-Agent Reinforcement Learning"
_TMLR — Under review for TMLR_

### Review · Reviewer_XZN7 · 2025-12-31

**Summary Of Contributions:**

This work proposes SHEP, a mesoscopic guidance framework for large-scale MARL that prioritizes experience replay using spatial heterogeneity descriptors (occupancy entropy, action diversity entropy, and Moran’s I). The key idea is to map high-dimensional microscopic agent states into a low-dimensional, interpretable group-level feature space, and to use this space to drive prioritized replay and a group-level variant of HER. The framework is positioned as a plug-in compatible with CTDE algorithms (notably MAPPO). Experiments on MAgent and SMAC demonstrate substantial gains in convergence speed and final performance.

**Audience:**

Yes

**Audience Explanation:**

The paper would be of interest to a meaningful subset of the TMLR audience, particularly researchers working on multi-agent reinforcement learning, scalability, experience replay, and coordination in large agent populations. The idea of using mesoscopic, physically interpretable spatial statistics to guide experience prioritization is likely to resonate with readers concerned with sample efficiency and non-stationarity in large-scale MARL.

**Broader Impact Concerns:**

There is no significant broader impact concerns. The paper is methodological and focused on improving learning efficiency in MARL. Potential applications are standard for MARL research, and no obvious ethical, societal, or safety risks are introduced beyond those already present in existing multi-agent systems research.

**Claims And Evidence:**

Yes

**Claims Explanation:**

The main empirical claims (e.g., improved sample efficiency, faster convergence, and stronger performance) in large-scale MARL settings are supported by clear experimental evidence on MAgent and SMAC. The reported results are consistent across multiple maps and scales, and the advantages over MAPPO, QMIX, and MFAC are convincing for me.

**Requested Changes:**

I would like to request changes on the following points:

1. Please include more comprehensive ablation studies to isolate the contribution of each component.

2. highlight the novelty of proposed method relative to existing PER/HER-based MARL methods.

4. I would like to see more discussion on the computational overhead to support scalability claims.

5. Further discussion to better justify broad claims about dimensionality and generality.

---

### Review · Reviewer_EfMJ · 2026-02-01

**Summary Of Contributions:**

The paper introduces a replay prioritisation framework for large scale MARL that operates at a mesoscopic level rather than on individual agents or full joint states. It summarises collective behaviour using spatial heterogeneity statistics obtained by projecting agents into macro cells over a metric space and computing measures of spatial coverage, action diversity, and structural coordination. These measures are combined into a group level score to identify critical moments and prioritise trajectory segments for replay. The framework augments standard training with group based replay weighting, auxiliary intrinsic rewards, and hindsight relabelling, while remaining compatible in implementation with centralised training and decentralised execution methods. Experiments on spatial multi agent benchmarks show improved learning efficiency and performance in large scale adversarial settings.

**Additional Comments:**

Under what precise conditions are the assumptions about temporal correlation and macroscopic dominance expected to hold, and how does the method behave when these conditions are violated?

Does the number of agents vary within episodes, across episodes, or not at all in the experiments, and how is the centralised critic defined in the presence of such variation?

How sensitive is performance to the choice of metric, macro cell resolution, and weighting of the heterogeneity terms? Which components of the framework are essential for the observed gains, and which are auxiliary?

**Audience:**

Yes

**Audience Explanation:**

The paper tackles an important scalability issue in MARL by shifting attention from microscopic agent level signals to mesoscopic group level structure.

The construction of spatial heterogeneity descriptors is intuitive in physically grounded environments and aligns well with the empirical domains considered.

The proposed prioritisation mechanism is simple to implement and integrates naturally with existing replay based MARL pipelines.

The empirical results on large scale spatial benchmarks are strong and suggest that replay guided by collective structure can accelerate learning and improve coordination in adversarial settings.

**Claims And Evidence:**

No

**Claims Explanation:**

Some of the constructs and ideas in the paper are underpinned by statements that are presented as a universal laws when in fact they are simply regimes that are common in some subsettings. For example the authors state “Multi-agent tasks have strong temporal correlation, and critical moments usually span several continuous steps (e.g., engagement sequences)” without substantiation.  For example this would not hold in repeated stage games e.g. the repeated prisoner’s dilemma. Even arguably such a property may not hold for MFGs.

Another example of this looseness is when the author say “In large-scale group tasks, the marginal contribution of a single agent’s microscopic action to the global outcome is significantly reduced, while the macroscopic behavioral patterns of the group become the key factors determining task progression.”

These claims undermine the paper and, for technical rigour should by qualified with conditions as to when these properties can be expected to hold. If the methods presented in the paper depend on such properties then the conditions that lead to such properties should be clearly stated.

Page 4 - “In large-scale MAS, the number of agents N often varies dynamically”. The authors would like to consider a setting in which the number of agents is allowed to vary within a given problem setting. This setup is not compatible with the Dec-MDP formalism on which the paper is based.  Also, we can’t expect standard MARL/CT-DE training regiments (such as MAPPO) to which the authors say their method can be attached to be able to handle such a setting (the critic in such a case will be ill-defined). In any case, it then requires one to disentangle where there are dependencies on the agents being fixed to determine whether the results of the paper at all rely on this restriction or not. Statements such as “Regardless of how the number of agents N fluctuates in the environment” suggest the authors genuinely want consider a setting with a varying number of agents). Seemingly the authors are conflating “scale-invariant mesoscopic features” with “solving a variable-$N$ Dec-POMDP using standard CTDE which is an error.

To me the paper seems to overreach in some places and the framework is presented as being more general than is the case. While it’s objectives in wanting to go beyond highly contextualised hierarchical mechanisms, it seems that the framework devises solutions that suffer the same fate but the restrictions to particular domains are simply unacknowledged. For example, many of the tools and methods the author seems to rely on the problem being defined on a spatial environment (i.e. those that are physical tasks with some underlying Euclidean geometry). This doesn’t seem to be fairly stated as a restriction in the setting (though it is mentioned as a part of the method). Also, objects like the relative advantage which “quantifies the dominance our group relative to the opponent” only make sense in certain problem settings (one condition being there being a clear opponent group). In fact this is slightly at odds with the stated problem setting which is cooperative.

The authors highlight unbiasedness in their IS correction for prioritised sampling but then they also add intrinsic rewards and Group-HER relabelling. The authors then make no attempt to provide evidence that the goal of unbiasedness is actually attained. Only under some conditions is unbiasedness achieved with these modification of the objective e.g. potential based reward shaping [6].
The stated goal of the paper is to construct a hierarchical framework which is general and doesn’t suffer the limitations of existing methods that are restricted to particular settings. However, there are design choices which are inherently limiting (and seem quite ad-hoc). For example,  the notion of a metric is for the density field projection mechanism. In physical settings this is fine because there is a clear notion of distance but how would one go about constructing a relevant metric in for example, a problem of a set of economic actors in bargaining tasks or language-based coordination? And I’m sure there are other examples better than this where such a notion would be meaningless.

Although the limitations of existing work are well documented, the literature review seems to contain few references for each topic (for example, the MFG literature is addressed only with 1-2 references). Without a more exhaustive account of the existing works, it’s hard to know whether the claims made fully account for the current state of affairs. I also feel that very relevant works in hierarchical MARL have been missed e.g. [1, 2]. There are also other hierarchical guidance mechanisms that I think are relevant here e.g. [3,4] and works that consider groupings e.g. [5]. All of these latter works point to subtopics that seem not to have been covered but I think are relevant to this line of inquiry.

Many of the mechanisms on the paper are constructed using linear combination of statistics whose properties and utility have been individually studied within the paper. However the paper makes no attempt to examine the effect of combining these objects such as correlations. For example, the authors define a comprehensive group score $E(t)$ which consists of terms that measure spatial coverage, action diversity, and structural coordination. However, correlations between these terms don’t seem to be studied. This is an important but missing element since these terms may interact and cause overcounting or biases in the measure $E(t)$). A similar observation applies to the unnormalised priority score (for example, it seems conceivable that a sharp, unexpected change in group position could lead to an increase in all 3) but it’s not clear whether such correlations would be helpful or detrimental.
The paper introduces quantities that provide some measure of spatial coverage, action diversity, and structural coordination. Though the paper does a good job of explaining these concepts within their physical concepts, I think the paper would benefit from examples within the multi-agent context that provides a detailed explanation of the benefits of collecting these statistics in a given scenario. I would also like to see some attempt to explain the generality of the usefulness of these statistics, for example, by providing two or more examples drawn with very different characteristics.

[1] Yang, Mingyu, et al. "Hierarchical multi-agent skill discovery." Advances in Neural Information Processing Systems 36 (2023): 61759-61776.
[2] Xu, Zhiwei, et al. "Haven: Hierarchical cooperative multi-agent reinforcement learning with dual coordination mechanism." Proceedings of the AAAI conference on artificial intelligence. Vol. 37. No. 10. 2023.
[3] Mguni, David Henry, et al. "MANSA: learning fast and slow in multi-agent systems." International Conference on Machine Learning. PMLR, 2023.
[4] Mguni, David, et al. "Coordinating the crowd: Inducing desirable equilibria in non-cooperative systems." arXiv preprint arXiv:1901.10923 (2019).
[5] Li, Mengxian, Qi Wang, and Yongjun Xu. "Gtde: Grouped training with decentralized execution for multi-agent actor-critic." Proceedings of the AAAI Conference on Artificial Intelligence. Vol. 39. No. 17. 2025.
[6] Ng, Andrew Y., Daishi Harada, and Stuart Russell. "Policy invariance under reward transformations: Theory and application to reward shaping." Icml. Vol. 99. 1999.

**Requested Changes:**

1. Please qualify claims that are currently presented as universal. Statements about strong temporal correlation, multi step critical moments, and macroscopic dominance do not hold for many multi agent settings. Clearly state the environmental conditions under which these assumptions are expected to apply, and revise the language accordingly.

2. Clarify the treatment of the number of agents. The claim that the agent count varies dynamically is not compatible with the Dec POMDP formalism used in the paper. Specify whether this refers to inactive agents or true population changes, and explain how the learning setup and centralised critic remain well defined. Revise the text if such variation is not actually evaluated.

3. Narrow the scope of the framework’s claimed generality. The method relies on a meaningful metric space, spatial partitioning, and opponent relative group scores, which restrict applicability to spatial and typically adversarial settings. These should be stated explicitly as assumptions rather than implied universality.

4. Add ablations and diagnostics. Analyse correlations between the heterogeneity statistics, sensitivity to macro cell resolution, and isolate the contribution of prioritisation, intrinsic rewards, and group based relabelling.

5. Strengthen the empirical and literature support. Include experiments or discussion of settings where the assumptions may fail, and expand the related work to better situate the contribution within hierarchical and group level MARL.

---

### Review · Reviewer_cPkK · 2026-04-16

**Summary Of Contributions:**

The paper aims to address the curse of dimensionality in multi-agent reinforcement learning systems, where the joint state-action space grows exponentially with respect to the number of agents.
To do so the paper focuses on identifying important samples and uses a prioritized experience replay.
Specifically, the paper proposes a mechanism called SHEP that scores a sample based on (1) occupancy diversity of the agents, (2) action diversity, and (3) topology formed by the location of the agents.
This score is used to define the sample priority along with the temporal-difference error priority, defining intrinsic reward for the agents, and constructing hindsight experience.
The paper presents experimental results evaluated on MAgent and SMAC environments, and compared against MAPPO, QMIX, and MFAC.
The results suggest that SHEP consistently outperforms all baselines under different environmental setup.

At this stage of the paper I recommend reject.

### Comments
- The writing is poor:
  - A lot of missing definitions or unexplained quantities, most of which relates to the fact that the paper provides little to no background around the algorithms this paper builds upon, e.g., PER, TD-learning.
    - In section 1, last paragraph, CTDE has not been introduced.
    - In the decentralized POMDP, it is not explained how the observations $o_t^i \in \mathcal{O}$ is generated.
    - What is $n_i(t)$? What is $i$?
    - The (2) in the two levels of the projection mechanism described in page 5 is unclear---what is normalized in the representation?
    - Action diversity entropy is never propoerly defined.
    - $E_\mathrm{self}$ and $E_\mathrm{opp}$ are never defined---what are $\mathrm{self}$ and $\mathrm{opp}$ in this case?
    - What's $A$? Isn't $I(t)$ already included in $E$ which is part of $\Delta \delta(t)$?
    - The expectation of the trajectory-level priority ranking should be clarified---my understanding is that it is empirical rather than true expectation.
    - What is $\nabla L_i$?
  - The paper briefly discusses the Moran's I quantity, e.g. intuition of $I(t) > 0$ and $I(t) \approx 0$, but many of the aspects are missing. For example, what happens if $I(t)$ is negative? At what scale does $I(t) > 0$ become significant, e.g. $> 1e-1, >1e1, > 1e2$, etc.?
  - The paper does not explain the newly-defined intrinsic rewards or the "group hindsight experience replay" in detail.
- The paper has little to no support to many claims:
  - One of the main point in the methodology is "unbiased-weight optimization", which I believe was originally addressed in PER. Furthermore, it would be worthwhile to discuss the difference between the trajectory-wise priority and prioritized sequence experience replay [1].
  - While not directly indicated, my interpretation of this paper suggests that existing methods are domain dependent while SHEP is not---if that is true I fully disagree because this depends on the definition of the macro-cells and the metric space. Whether or not this is "easier" to define is not discussed at all.
  - The last sentence of page 7 gives a high-level intuition of what $\delta(t)$ should capture, but there is no evidence at all to support this.
  - The paper provides no hyperparameter tuning nor sensitivity analysis over the newly introduced hyperparameters. Nor are there any ablations on the importance of each introduced components.
  - The paper provides only performance curve of the agents, and speculates why SHEP outperforms existing methods without any further analysis. For example, in open area the paper claims SHEP "avoids high-dimensional meaningless random talks" but never explores any metrics that may support this, e.g. trajectory diversity of SHEP over training.

### References
- [1]: Brittain, M., Bertram, J., Yang, X., & Wei, P. (2019). Prioritized sequence experience replay. arXiv preprint arXiv:1905.12726.

**Audience:**

Yes

**Audience Explanation:**

I believe the topic is of interests for some TMLR's audience, perhaps the audience can benefit from knowing the existence of a method that outperforms the compared baselines---however, there are limited findings in the paper and most of the claims are not well supported, as mentioned above.

**Claims And Evidence:**

No

**Claims Explanation:**

See above.

**Requested Changes:**

Generally, the paper can greatly from both writing and extra analyses:
- Properly define the formulations and terms in the paper, especially the "introduced" intrinsic reward and grouped HER mechanisms.
- A proper ablation study on the impact of each introduced mechanism and hyperparameter sweep both SHEP and the baselines.
- Extra analysis to support behavioural claims, such as "narrowing search space".